# The Effectiveness of Learning to Use HMD-Based VR Technologies on Nursing Students: Chemoport Insertion Surgery

**DOI:** 10.3390/ijerph19084823

**Published:** 2022-04-15

**Authors:** Ae-Ri Jung, Eun-A Park

**Affiliations:** 1College of Nursing, Bucheon University, Bucheon 14774, Korea; aeri@eulji.ac.kr; 2College of Nursing, Eulji University, Uijeongbu 11759, Korea

**Keywords:** nursing education, nursing student, simulation, virtual reality

## Abstract

Background: The purpose of this study was to develop a mobile head mounted display (HMD)-based virtual reality (VR) nursing education program (VRP), and to evaluate the effects on knowledge, learning attitude, satisfaction with self-practice, and learning motivation in nursing students. Methods: This was a quasi-experimental study using a nonequivalent control group pretest-posttest design to evaluate the effects of HMD-based VRP on nursing students. A Chemoport insertion surgery nursing scenario was developed with HMD-based VRP. The experimental group consisting of 30 nursing students underwent pre-debriefing, followed by VRP using HMD and debriefing. The control group, consisting of 30 nursing students, underwent pre-debriefing, followed by self-learning using handouts about Chemoport insertion surgery procedures for 30 min, and debriefing. Results: The experimental group that underwent HMD-based VRP showed significantly improved post-intervention knowledge on operating nursing (*p* = 0.001), learning attitude (*p* = 0.002), and satisfaction (*p* = 0.017) compared to the control group. Sub-domains of motivation, attention (*p* < 0.05), and relevance (*p* < 0.05) were significantly different between the two groups, post-intervention. Conclusions: HMD-based VRP of Chemoport insertion surgery is expected to contribute to knowledge, learning attitude, satisfaction, attention, and relevance in nursing students.

## 1. Introduction

The goal of nursing education is to promote the application of theoretical knowledge in clinical practice [1]. Nursing students are needed to improve their problem-solving and clinical reasoning capabilities because higher problem-solving skills can provide high-quality care in clinical nursing practice [2]. However, limited clinical practice time affects the opportunity for students to have clinical experience with real patients [3]. Narrowing the gap between theory and practice during the educational process is necessary [4]. As the severity and emergency of patients are becoming diversified, hospitals are demanding graduate nurses with critical thinking and clinical judgment capable of performing complex roles with a certain level of clinical performance [5]. Such demands and limitations of clinical practice education bring several challenges to the nursing educators

In nursing education, a simulation that integrates clinical practice and theoretical learning reproduces reality, facilitates active learning participation, and provides opportunities for repetition, feedback, evaluation, and reflection and has been used as the main teaching-learning strategy to improve the competency of nursing students [6]. Simulation artificially reproduces scenarios with an educational tool or technique by manipulating or applying a stimulator to education and training [7]. This allows the students to think in the reproduced reality that is similar to the actual situations [8].

Simulation education improves integrated and critical clinical judgment and problem-solving skills and enables the link between knowledge and skills in students [9]. In many previous studies, simulation education was effective for the management of emergency situations, which are rare experiences in clinical practice for nursing students [9,10]. Additionally, compared to the conventional education method, simulation education significantly improved the academic achievement of nursing students [11]. Clinical practice is associated with a high risk of infection and a lack of patient privacy. In contrast, simulation education provides the opportunity for students to experience the reproduced reality without posing any threat to the safety of patients [12], reduces stress for students [13], and offers repeated learning without fear of mistakes in safely reproduced reality [13]. Furthermore, we can see the application of the latest virtual reality (VR) technology even in the highly sophisticated and difficult neurosurgery area recently; AR neurosurgical navigation platform provides an unprecedented 3D visualization of both the surgical field and virtual elements, improves the depth-perception of the augmented scene, and proved to be capable of effectively supporting the surgeon in performing highly precise and accurate craniotomies [14]. VR body swapping experiments using visuo-tactile stimulation have reached the level of consideration to use as an intervention for patients suffering from eating and weight disorder [15]. 

Simulation education using highly realistic mannequins can be applied to only a small number of students at a time and requires a separate space as well, and there are high costs for the installation, management, and repair of equipment [16]. Additionally, this simulation education method is heavily affected by the instructor’s level of competency and offers limited interaction with patients [17]. To compensate for such limitations, a nursing simulation program using VR has been introduced. VR-based programming is highly preferred for its greater effectiveness, improvement of concentration in students, and customized education compared to the conventional education method [18].

VR is a computer simulation that promotes student learning in a realistic three-dimensional clinical environment without jeopardizing patient safety. A 3D virtual world for education provides convenient access to information and actively utilizes information during learning to increase the user’s motivation to learn [19]. In comparison with the control group, there were significant differences between increase in knowledge, maintenance and stability of knowledge, dissemination of knowledge, cognitive, anxiety, change of attitude and emotional connection, relief stability, satisfaction, presence, performance, skill, interaction, and decision-making [20].

Operating room nursing requires a high level of skills from routine surgical procedures to emergencies [21], constant tension to maintain smooth interpersonal relationships, and the skillful use of various devices and specialized equipment, as well as quick and agile actions [22]. However, the high level of knowledge required for different equipment and consumables in operating rooms and the complicated surgical procedures often creates fear for nursing students and a feeling of being powerless [23]. As a result, various educational methods are required for the practical education of nursing students. Therefore, this study aimed to develop and evaluate the effectiveness of simulation programs for operating room nursing, providing a rare nursing experience for only a limited number of students during clinical practice.

The purpose of this study was to develop head mounted display (HMD)-based VRP and understand the effects of the program on the knowledge, learning attitude, practice satisfaction, and learning motivation of nursing students.

## 2. Materials and Methods

### 2.1. Study Design

This was a quasi-experimental study using a nonequivalent control group pretest-posttest design to develop and evaluate the effects of HMD-based VRP of Chemoport insertion surgery for nursing students. The recruitment and intervention of participants was from 1 April to 30 June 2021.

### 2.2. Study Participants

The participants of this study were second-year students, third-year students, and fourth-year students in the Department of Nursing at B University in Gyeonggi-do, who understood the purpose and procedure of the study and voluntarily agreed to participate. A notice was posted to recruit participants through the department bulletin board. Participants who had never experienced HMD-based VRP were recruited, considering that the age distribution of nursing students was diverse regardless of grade. The participants were separated into experimental and control groups. The minimum number of participants required for this study was calculated using G*Power 3.1.9.7 with a power (1-β) of 0.80, a significance level (⍺) of 0.05, and an effect size of 0.66, as in a previous study [22]. All 60 participants who were recruited were included in the final study. A total of 30 participants was required for each group. No one was eliminated during the study.

### 2.3. Tools

In this study, structured questionnaires (in Appendix A) were used to assess knowledge, learning attitude, satisfaction with self-practice, and learning motivation.

#### 2.3.1. Knowledge

To evaluate pre- and post-operative nursing knowledge, 10 items on the key knowledge of Chemoport insertion surgery nursing were developed. The content validity of the items was verified by two nursing professors and one expert with a doctorate degree in nursing and more than 10 years of clinical experience. The content validity index (CVI) was 0.94, suggesting good validity of content for each questionnaire (in Appendix A) item. Correct and incorrect answers were scored either one or zero for each item, respectively, and the total score of the questionnaire (in Appendix A) ranged from zero to 10 points. A higher score indicated greater pre- and post-operative nursing knowledge. Cronbach’s α for the questionnaire (in Appendix A) was 0.73.

#### 2.3.2. Learning Attitude

Learning attitude was evaluated using a tool for the assessment of attitudes, habits, beliefs, and motives of students in class, developed by the Korea Educational Development Institute and modified by Hwang [23]. The tool consists of 16 items on self-concept, study attitude, and learning habits. The items were evaluated on a five-point Likert scale, from one point for “not at all” to five points for “always”. The total score ranged from 10 to 80 points, and a higher score indicated a greater learning attitude. The reliability of the tool was 0.84 in a previous study [24] and 0.71 in this study.

#### 2.3.3. Satisfaction with Self-Practice

Satisfaction with self-practice was evaluated using a tool developed by Yoo [25] and modified according to the purpose of this study. The tool originally consisted of 24 items at the time of development. In this study, the number of items was reduced to 17. Two nursing professors evaluated the content validity of the items. The CVI was 0.95. The tool consisted of the following domains; learner attitude, learner satisfaction, appropriateness of learning content, learning achievement, motivation, debriefing, and self-reflection. Each item was evaluated on a five-point Likert scale from one point for “not at all” to five points for “always”. The total score ranged from 17 to 85 points, and a higher score indicated greater satisfaction with self-practice. Cronbach’s α was 0.94 in the study by Yoo [25] and 0.73 in this study.

#### 2.3.4. Learning Motivation

Learning motivation was evaluated using a tool based on the Instructional Materials Motivation Scale (IMMS) by Keller [26], which was modified by Jang [27]. IMMS was developed to measure motivation-related problems in the self-directed learning of students, and validation was proven in mixed reality instructional simulation [28]. The tool consisted of 34 items in total, including a four-dimensional measurement tool (12 items on attention, 9 items on relevance, 8 items on confidence, and 5 items on satisfaction). Each item was evaluated on a five-point Likert scale from one point for “not at all” to five points for “always”. The total score ranged from 34 to 170 points, and a higher score indicated greater learning motivation. Cronbach’s α was 0.96 in the study by Jang [27] and 0.67 in this study.

### 2.4. Intervention

#### 2.4.1. Intervention Development Process

Step 1: First, the overall scenario was constructed through a literature review for content development. Second, one surgeon and one surgical nurse with extensive experience in Chemoport insertion surgery were interviewed for information on Chemoport insertion surgery indications, surgical procedures, necessary supplies and equipment, essential knowledge, and precautions. Third, the angiography room and surgical equipment, instruments, and items were observed. The surgery was observed several times to complete the scenario.Step 2: After creating a storyboard and scenario for Chemoport insertion, the contents were reviewed by the surgeon and operating nurse for revision.Step 3: Resource 3D modeling (using Autodesk 3DS Max for 3D modeling and rendering, and Pixologic ZBrush for sculpting and 3D modeling) and voice scripts were developed.Step 4: Each scene was created with 3D animation (3DS Max).Step 5: VR rendering (using the Arnold renderer in 3DS Max) was conducted for each scene.Step 6: After organizing the scenes in an appropriate order, the audio script and video were matched (using Adobe After Effects for video and effect production).Step 7: After checking the development contents, usability was verified by three operating nurses working in tertiary general hospitals with more than 10 years of experience. Ease of use and student satisfaction were evaluated, and the final revision of the contents was conducted based on the feedback.Step 8: The HMD-based VRP was applied to nursing students and evaluated. It proceeded in stages of pre-briefing, learning, debriefing, and pre-post survey.

#### 2.4.2. Intervention Application and Data Collection

Data were collected from 60 nursing students from April to June 2021. Students who understood the purpose of the study and voluntarily agreed to participate were enrolled. The participants were blinded so that they did not know their groups. The participants completed a preliminary questionnaire (in Appendix A) after signing the consent form for study participation. Any students with a major mental disorder and taking psychopharmacologic drugs were excluded from the study for possible risk of dizziness from the intervention.

The HMD-based VRP was provided to the experimental group of 30 participants. Pre-briefing was provided first, followed by VR simulation programs and debriefing. During the pre-briefing, the participants were educated on the program content, how to use the HMD and the precautions. Before learning, orientation on the patient’s situation and the scenario was given to the participants. In addition, the students were given enough time to touch and learn the functions so that they could become familiar with the HMD device. The participants underwent the intervention one by one. The program content application was executed, and the HMD was worn. An angiography room was reproduced virtually, and the surgical process was observed. The participants were allowed to control the VR and see the angiography room and Chemoport insertion surgery process in 360 degrees. Self-learning was conducted for approximately 30 min on a chair for the safety of the participants. The researcher monitored the entire process of intervention of each participant, and Q&A and debriefing were conducted afterward. After the students finished learning the program, the researcher provided reflection, discussion, and feedback on the situation in which they experienced the program during the debriefing process (Figure 1).

The control group underwent orientation on the contents of Chemoport insertion surgery provided by an operating nursing instructor, followed by 30 min of self-learning using a handout on Chemoport insertion surgery. Debriefing was conducted with the chief investigator, and any questions by the participants were answered.

### 2.5. Ethical Considerations

This study was approved by the National Institutional Review Board (IRB No. P01-202103-11-001). Prior to the study, the participants were informed of the study’s purpose and were assured of anonymity and confidentiality. The participants were also told they could withdraw their consent at any time. Those who voluntarily agreed to participate provided a written consent form and completed the questionnaires (in Appendix A). For the control group, HMD-based VRPs were provided to those who wished to undergo the education after completing the intervention. As the participants were students in the affiliated institution of the researchers. Thus, concerns about disadvantages due to refusal to participate could affect the decision for voluntary participation. The participants were informed that there was no compulsory participation and that they will not have any disadvantage from their refusal to participate in the study. Additionally, both the recruitment and consent forms included explanations that the participants could participate, withdraw, or suspend their participation at any time, for any reason.

### 2.6. Data Analysis

The collected data were analyzed using IBM SPSS Statistics for Windows, ver. 22.0 (IBM Corp., Armonk, NY, USA). Frequency/percentage and mean/standard deviation were calculated for the general characteristics and each variable of participants. Normality was tested. T-test and chi-square tests were conducted to assess the homogeneity of the participants. A paired t-test was conducted to evaluate the effects of interventions in control and experimental groups. The level of significance was set to *p* < 0.05. Cronbach’s ⍺ was evaluated to analyze the reliability of the evaluation tools.

## 3. Results

### 3.1. Participant Characteristics and Baseline Test of Homogeneity

A final total of 60 participants were included in the study, with 30 participants in each of the groups, control and experimental. The baseline test of homogeneity for general characteristics and study variables of the two groups showed no significant difference, suggesting homogeneity between the two groups. The results are shown in Table 1.

### 3.2. HMD-Based VRP Effect Evaluation

Table 2 shows the effects of the HMD-based VRP on the knowledge, attitude, satisfaction, and motivation of the experimental and control groups.

#### 3.2.1. Knowledge

In the experimental group, the score for knowledge out of 10 points increased from 5.37 ± 1.13 points pre-intervention to 6.97 ± 1.35 points post-intervention. In contrast, in the control group, the score decreased from 5.13 ± 1.57 points before the intervention to 4.80 ± 1.65 points after the intervention. There was a significant difference in the score for knowledge between the two groups (*t* = 4.01, *p* = 0.001).

#### 3.2.2. Learning Attitude

In the experimental group, the score for attitude out of 80 points increased from 55.03 ± 5.48 points pre-intervention to 60.00 ± 6.94 points post-intervention. In the control group, the score decreased from 53.03 ± 6.70 points before intervention to 50.87 ± 8.03 points after intervention. There was a significant difference in the attitude score between the two groups (*t* = 3.25, *p* = 0.002).

#### 3.2.3. Satisfaction with Self-Practice

In the experimental group, the satisfaction scores out of 85 points increased from 69.70 ± 11.99 points pre-intervention to 75.00 ± 10.49 points post-intervention. In the control group, the score decreased from 65.47 ± 12.50 points before intervention to 64.17 ± 14.31 points after intervention. There was a significant difference in the satisfaction score between the two groups (*t* = 2.46, *p* = 0.017).

#### 3.2.4. Learning Motivation

In the experimental group, the motivation scores out of 170 points increased from 111.37 ± 18.21 points pre-intervention to 118.37 ± 16.86 points post-intervention. In the control group, the score increased from 99.57 ± 17.89 points before intervention to 100.30 ± 23.51 points post-intervention. There was no significant difference in the total score for motivation between the two groups (*t* = 1.59, *p* = 0.118). However, in the sub-domains, there were significant differences in attention (*t* = 2.51, *p* = 0.016) and relevance (*t* = 2.10, *p* = 0.040) between the two groups. Table 2 shows the effects of the HMD-Based VRP on the knowledge, attitude, satisfaction, and motivation of the experimental and control groups.

### 3.3. The Instructor-Led Debriefing

Debriefing was conducted individually in the experimental group. As a result of analyzing the responses of the students in the experimental group during debriefing, the following answers were provided. “I was able to actually feel like I was working in a clinical practice”, “It stimulated interest in learning, increased concentration, and helped me to learn achievement”, “I was able to learn a sense of realism while experiencing it”, “It was memorable because it took less time than actually going to practice and I was able to learn over and over again”, and “I would like to perform the technique myself for higher learning outcomes”. On the other hand, the students who participated in the control group answered as follows; “The surgery terms are difficult and unfamiliar to me, so it is difficult to memorize it” and “I had never seen the procedure in the surgery room, so it was not easy to understand”.

## 4. Discussion

The intervention program in this study was executed using mobile HMD-based VR. Smartphones were inserted into the Distributed Interactive Virtual Environment of the mobile HMD to execute the VR-based Chemoport nursing program. Wireless HMD functions both as a monitor and a computer. Wireless HMD is relatively cheap and is not temporally and spatially restricted. However, it is difficult for the instructor to observe the learner’s progress and provide immediate feedback [29]. In this study, the VR contents were developed as a 360-degree panoramic VR. The participants used a mobile HMD and observed the operation room, surgical instruments, and surgical scenes in a 3D view from a third-person perspective. 3D simulation, YouTube, and 360-degree web-based VR learning are also widely used technologies for students’ learning, but they differ from the developed technologies in terms of presence. Presence has been defined as “being in a normal state of consciousness and having the experience of being inside a virtual environment” [30]. Compared to YouTube or web-based VR, which are screen-based environments, the outside is blocked from view, and the display is exposed very closely to the user’s view at a location, so it gives a higher sense of presence and immersion, and shows a higher learning effect [31,32,33].

The score for learning content-related knowledge increased from 5.37 ± 1.13 points to 6.97 ± 1.35 points in the experimental group and decreased from 5.13 ± 1.57 points to 4.80 ± 1.65 points in the control group. The post-intervention score was significantly different between the two groups. Consistent with our results, previous studies [34,35] and meta-analysis [35,36] also reported that VR was effective in improving knowledge. There were significant differences in the response rate of items on ultrasound devices used to find the anatomical insertion site of Chemoport and veins between the two groups. The students are often unfamiliar with complex surgical techniques, surgical instruments/devices, and terms. Using VR for learning increases the sense of spatial presence. Additionally, VR training is effective in delivering procedural knowledge [35], which is thought to have enhanced the learning effects of operating nursing in the experimental group.

Our findings showed that the learning attitude score out of 80 points increased from 55.03 ± 5.48 points pre-intervention to 60.00 ± 6.94 points post-intervention in the experimental group. In the control group, the score decreased from 53.03 ± 6.70 points pre-intervention to 50.87 ± 8.03 points post-intervention. The post-intervention score for learning attitude was significantly different between the two groups. This observation agreed with that of a previous study [37] in which learning attitude increased after simulated scenarios. These findings suggested that VRP experiences induce interest and motivation to learn in students, leading to active participation.

Satisfaction with self-practice out of 85 points increased from 69.70 ± 11.99 points pre-intervention to 75.00 ± 10.49 points post-intervention in the experimental group. In the control group, the score decreased from 65.47 ± 12.50 points pre-intervention to 64.17 ± 14.31 points post-intervention. The post-intervention score for satisfaction with self-practice was significantly different between the two groups. Consistent with our finding, a previous study [38] reported that VR increases the satisfaction of users. Altogether, these findings suggest the necessity of using content based on VR or augmented reality for practical education.

Herein, both experimental and control groups showed increased learning motivation post-intervention; however, there was no significant difference between the two groups. This finding contrasted with that of a previous study [39], in which VR learning increased motivation. However, there were differences in the subdomains, attention, and relevance between the two groups, which agreed with a previous study on using VR-based learning for the improvement of students’ attention [40]. In our study, the students were unable to directly perform the surgical techniques due to the limitations of VR education from the third-person perspective. It is believed that it provides only limited improvement in students’ self-confidence in their skills and their satisfaction with their motivation. This finding is partially in agreement with the results of a meta-analysis by Chen et al. [36] on VR studies in nursing education. In contrast, user evaluation by Mao et al. [41] on immersive virtual reality shows good satisfaction and low discomfort. Therefore, follow-up studies must be conducted to confirm the effects.

VRP development in this study was conducted in the early stage of rapid development of VR technology content production, in response to the increased demand for contactless practice due to the COVID-19 pandemic. In future studies, using a haptic device instead of simple observation would be necessary to allow the learner to participate in skill simulation directly and execute the skills of interest to improve the learning effect, engagement, and skill confidence, thereby increasing learning motivation [42].

VRP is expected to improve spatial awareness, experiential learning, engagement, context learning, and cooperative learning [43]. In this study, spatial awareness and experiential learning were used to experience things that are difficult to encounter in clinical practice, and visual and sensory interaction for easy engagement with the operating environment were provided through VR technology. Furthermore, virtual worlds and avatars, which were not provided in this study, would enable cooperative learning.

VR simulation content development has several limitations, such as technical issues and a lack of realism in building the virtual reality platform [44]. To develop and execute high-fidelity programs, multidisciplinary cooperation between professors, clinical experts, and IT technology experts would be required. In addition, improvement of the instructors’ competency, proper relocation of the clinical practice, provision of technical experts, and device technology development would minimize VR sickness [45,46], leading to enhanced execution of VR-based learning programs.

Instructors’ poor understanding of the skills required can increase the burden and fear of students in learning new methods. Therefore, instructors should be required to educate for easy access to and understanding of VR technology.

It was reported that the debriefing process is the most important component of a simulation-based learning experience [47]. Key elements of debriefing for educators to keep in mind include the following; approach, learning environment, engagement of learners, reaction, reflection, analysis, diagnosis, and application [48]. In this study, through debriefing, the response and reflection of VRP were discussed; realism, presence, concentration, interest, complaints of not being available for hands-on experience, and unfamiliarity. It was found that VRP learning provided students with higher interest and understanding of the situation than paper learning. However, in order to improve learning outcomes, it is necessary to improve problem-solving ability through the more perfect virtual space and direct actions. These are the elements that should be considered when developing an upgraded version of VRPs in the future.

## 5. Conclusions

This study is significant in that it evaluated the learning effects of HMD-based VRP among nursing students. However, several limitations should be considered in this study’s findings. First, the participants were nursing students recruited from a single university. Therefore, the findings cannot be generalized to all nursing students. Second, from a technology aspect, this VRP is a new condition of 360 desktop viewing and low-fidelity VRP, so it is required to develop the high-fidelity VRP to provide the opportunity for direct interaction with a virtual environment and objects. Lastly, the verification of the effect of the VRP itself in this study was not objectively measured, but it seems clear that it had a positive educational effect on nursing students. In the future, Further research is needed to verify the effectiveness of VRP and improve its performance. This study is significant in several aspects. First, nursing students can experience HMD-based VRP that reproduces the actual environment and procedures in the inaccessible operating room in clinical practice. Second, it might increase expectations for the continuous development and application of VR in the clinical practice area.

## Figures and Tables

**Figure 1 ijerph-19-04823-f001:**
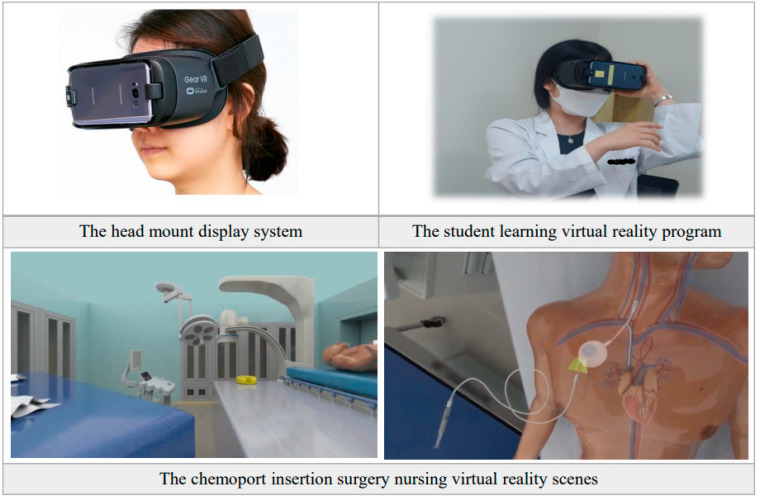
The virtual reality (VR)-based program in use.

**Table 1 ijerph-19-04823-t001:** Homogeneity Test of General Characteristics.

Characteristics	Categories	Total (*N* = 60)	Exp. (*N* = 30)	Cont. (*N* = 30)	*χ*^2^ or *t*	*p*
		N (%) or Mean ± SD				
Gender	Male	10 (16.7)	4 (13.3)	6 (20.0)	0.49	0.731
	Female	50 (83.3)	26 (86.7)	24 (80.0)		
Age		25.2 ± 6.45	23.10 ± 2.76	27.3 ± 8.25	21.79	0.150
Grades	4.0–4.5	11 (18.3)	5 (16.7)	6 (20.0)	2.33	0.507
	3.5–4.0	27 (45.0)	14 (46.7)	13 (43.3)		
	3.0–3.5	20 (33.3)	9 (30.0)	11 (36.7)		
	2.5–3.0	2 (3.3)	2 (6.7)	0		
Satisfaction of major in nursing	Very high	32 (53.3)	16 (53.3)	16 (53.3)	0.62	0.733
	High	18 (30.0)	8 (26.7)	10 (33.3)		
	Moderate	10 (16.7)	6 (20.0)	4 (13.3)		
Interesting of practice	Very high	32 (53.3)	17 (56.7)	15 (50.0)	1.97	0.374
	High	23 (38.3)	12 (40.0)	11 (36.7)		
	Moderate	5 (8.3)	1 (3.3)	4 (13.3)		
Knowledge		5.25 ± 1.36	5.37 ± 1.13	5.13 ± 1.57	5.83	0.443
Learning attitude		54.03 ± 6.15	55.03 ± 5.48	53.03 ± 6.70	28.64	0.155
Satisfaction		67.58 ± 12.33	69.70 ± 11.99	65.47 ± 12.50	24.67	0.646
Learning motivation		105.47 ± 18.86	111.37 ± 18.21	99.57 ± 17.89	43.00	0.386

Exp. = Experimental group, Con. = Control group, SD = Standard deviation.

**Table 2 ijerph-19-04823-t002:** Effects of a VR-based nursing simulation program.

Variables		Group	Pre Mean ± SD	Post Mean ± SD	*t(a)*	*p(a)*	*t(b)*	*p(b)*
Knowledge		Exp.	5.37 ± 1.13	6.97 ± 1.35	6.87	0.001 *	4.01	0.001
		Cont.	5.13 ± 1.57	4.80 ± 1.65	−0.79	0.878		
Attitude		Exp.	55.03 ± 5.48	60.00 ± 6.94	3.83	0.001 *	3.25	0.002
		Cont.	53.03 ± 6.70	50.87 ± 8.03	−1.22	0.231		
Satisfaction		Exp.	69.70 ± 11.99	75.00 ± 10.49	2.75	0.010 *	2.46	0.017
		Cont.	65.47 ± 12.50	64.17 ± 14.31	−0.69	0.493		
Motivation		Exp.	111.37 ± 18.21	118.37 ± 16.86	3.12	0.004 *	1.59	0.118
		Cont.	99.57 ± 17.89	100.30 ± 23.51	0.23	0.823		
Sub-domain	Attention	Exp.	40.53 ± 6.76	43.90 ± 7.24	4.34	0.000 *	2.51	0.016
		Cont.	37.77 ± 6.14	37.17 ± 8.37	−0.44	0.666		
	Relevance	Exp.	30.57 ± 5.93	33.47 ± 5.61	2.95	0.006 *	2.10	0.040
		Cont.	25.97 ± 5.98	25.87 ± 7.52	−0.10	0.924		
	Confidence	Exp.	22.33 ± 4.19	22.57 ± 2.79	0.39	0.698	−0.75	0.456
		Cont.	20.93 ± 4.43	21.90 ± 5.35	1.24	0.223		
	Satisfaction	Exp.	17.93 ± 3.40	18.63 ± 3.14	0.94	0.354	0.04	0.971
		Cont.	14.90 ± 3.59	15.37 ± 4.29	0.64	0.527		

*a*: Within a group, *b*: between groups, Exp. = Experimental group, Con. = Control group, SD = Standard Deviation, *: statistically significant.

## Data Availability

Data sharing not applicable.

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
