# Peer review of "The Effectiveness of Learning to Use HMD-Based VR Technologies on Nursing Students: Chemoport Insertion Surgery"

_ijerph, 2022, doi:10.3390/ijerph19084823_

Round 1
Reviewer 1 Report
In the intro you should state the importance of diversifying the teaching methods in nursing school, from the first-years to the final graduation
(https://www.mdpi.com/2076-3417/10/7/2357/htm; https://www.sciencedirect.com/science/article/pii/S1557308721000573; https://www.mdpi.com/1365606)
what do you mean with quasi-experimental study? please find a definition for Your research design
a copy of the questionnaires used should be included as well as a graphical representation of the results
Author Response
Thank you for your comments on the review. We learned more about this study while looking for answers to your review comments.
Please see the attachment

Reviewer 2 Report
Strengths
- The authors conducted a well designed study with appropriate consideration of power and item validation for evaluation instruments.
- The authors are examining how new technology could be integrated into existing curricula and how students could benefit from learning concepts in different modalities.
- The authors clearly motivate this work by highlighting scenarios and educational needs that benefit from lower-fidelity simulation (as opposed to mannequin simulations)
Weaknesses
- The main weakness is that the authors overstate the technological comparison between the two conditions. The authors attribute benefit to the "VR-based" condition, when the VR system is a minimally-interactive 360 video. The reported differences in results between conditions can not be attributed to "VR", since for example, much of the benefit might be realized from a desktop 360 viewer, such as in YouTube. There was no clear discussion or explanation how the higher level of immersion gained by used head tracking in a phone-based HMD would provide benefit.
- There are some needed statistical improvements including not summing Likert scales (while no full consensus on this, following current guidelines should be to avoid summing ordinal data). Also Likert questions means statements that are agree/disagree which 2.3.2 seems to point different labelling.
- The repetition of results in discussion was unnecessary. consolidate or streamline.
- As the authors pointed out, much work into the value of simulation in nursing education has already been done. The novelty here appears to be the specific scenario of Chemoport Insertion. That simulation was found beneficial for yet one more scenario does not significantly advance simulation research.
Recommendation
Overall this work studied how an HMD-based 360 video of a Chemoport Insertion surgery scenario would benefit nursing students. The authors are not able to identify what features of the intervention resulted in improvements in the metrics. The feature could be simply the 3D renderings, user controlled viewpoint, or the visual nature of the scenario. For example, a condition with just a desktop 360 video would identify what features were necessary to see improvement. That the user was a passive viewer of 360-based videos does not highlight the benefits of "VR" from a higher immersion perspective. That's one of the risks in doing comparisons between such different conditions (paper vs 3D simulation). Further, the contribution to the field of simulation and education is limited because numerous evaluative studies have identified similar benefits identified here, just the scenario of Chemoport insertion is new.
Minor:
- Vr-based (add hyphen)
- Through "the" department (missing article)
- Figure 1 uses HDM, should be HMD
- The authors should state if the educator or anyone that managed the students in the study blind to participation
- The authors should state if all participants (including the control participants) were given the "other" instructional experience after the study to enable balance in educational benefit.
Author Response

(The authors gave the same response as above.)

Reviewer 3 Report
Interesting topic, but paper needs some revisions:
- Lines 11-12 "This was a quasi-experimental study of nursing students" What do authors mean for quasi experimental? Please revise.
- Lines 48-51: "in contrast, simulation education provides the opportunity for students to experience the reproduced reality..." Improve these point, look at these papers: Augmented Reality-Assisted Craniotomy for Parasagittal and Convexity En Plaque Meningiomas and Custom-Made Cranio-Plasty: A Preliminary Laboratory Report. Int J Environ Res Public Health. 2021 Sep 22;18(19):9955. doi: 10.3390/ijerph18199955. -- Virtual Reality Body Swapping: A Tool for Modifying the Allocentric Memory of the Body. Cyberpsychol Behav Soc Netw. 2016 Feb;19(2):127.
- The mean age in the control group is higher than that in the experimental group. Could this represent a bias in the final results? if yes, please add this in limitations of the study seciton.
- Lines 64-67: "In previous studies, VR based nursing education significantly improved performance [11,17], satisfaction [11,17,18] , 65
knowledge [11,19]... " This can be interesting, but authors should show the role of "virtual reality" & "beyond planning" in 2022. Improve this point with refs. - Lines 304-305: "n. In that study, VR did not have significant effects on other areas such as knowledge and self-confidence" Good, however you should also report papere which show the opposite: ref. -- Immersive Virtual Reality for Surgical Training: A Systematic Review. J Surg Res. 2021 Dec;268:40-58.
- Line 319 "VR simulation content development has several limitations" then at lines 332-333 "However, several limitations must be considered in consideration of this study’s findings" Remove the last one. In the conclusion, authors should add what this paper add new to the literature.
- By reading the whole paper is not clear what authors think about VR and which are the results of this study. "VR was effective in improving knowledge" at line 271 and "VR did not have significant effects on other areas such as knowledge and self-confidence" at lines 304. Is VR useful or not useful? Revise.
- Lines 327-329: "Therefore, simple education to increase teachers’ understanding of VR technology and easy access to the VR technology would be required" this sentence does not make sense. Revise in English.
Author Response

(The authors gave the same response as above.)

Round 2
Reviewer 2 Report
In response to the concerns about overstating the technical comparison in the initial review, the authors aim to simply add clarification that their 360 video based HMD system has 1. higher presence and immersion than a desktop viewer and 2. higher sense of presence and immersion shows a higher learning effect in prior work. This doesn't address my concern which is it is not clear what technological innovation here lead to the improved performance in this study. To be sure, simpy using the increase immersion of HMDs have in some situations improve performance (as in the papers cited). However it is well understood that such improvement is scenario and skill specific, and not a blanket wide endorsement that all scenarios in higher sense of presence situations would lead to learning benefits. Thus simply clarifying that the HMD condition is different than a desktop viewer doesn't address the point. It is not clear what technological advance results in improvements. this would only be addressable with a new condition of 360 desktop viewing. Without the additional condition, then all that can be said is this scenario within a 360 video HMD resulted in benefit, but it isn't clear what the benefit came from. the implication that the benefit came from increased immersion (and probably sense of presence, but that wasn't measured and most measurements won't work between vastly different technological interventions) is not supported by the conducted study. Thus either the authors add the condition, or change significantly the claims of what the benefits and what the benefits can be attributed to.
Given that I still have methodological issues with the design, i unfortunately reach the same conclusion that the contributions of this work is limited since there is little understanding what caused the improvements in measures. Simply agreeing with the suggestion and doing light rewording is insufficient.
The significant of the study is overstated. That nursing students experienced a VR simulation (and simulation here is used quite liberally since simply replying 360 video is hardly a simulation) is not new or novel. this has been done in medical simulations for decades. The second significance is vague and unclear.
For other review elements, the authors did do a good job.
Author Response
Thank you for your in-depth advice. Please see the attached file for the answer.

Reviewer 3 Report
Authors solved all my criticisms.
Author Response
Thank you for your feedback. I'll do my best to become a better researcher.
Round 3
Reviewer 2 Report
The authors reduced the scope of their claims. The grammar/writing should go through review (I recommend the authors use a program such as grammarly to help before any subsequent drafts). The text needs quite a bit of cleaning up as the responses to rounds of reviewer issues has made some sections disjointed. otherwise, i think that overall the claims are modest, but substantiated by the study.
Author Response
Thank you for your answer and suggestions. The Grammarly program was very useful. We revised grammar, spelling, and writing. In addition, we received helping the editing program by editage korea.
